# Probiotics’ Efficacy in Preventing Asthmatic Allergic Reaction Induced by Air Particles: An Animal Study

**DOI:** 10.3390/nu14245219

**Published:** 2022-12-07

**Authors:** Chi-Yu Yang, Fang-Yu Zhang, I-Jen Wang

**Affiliations:** 1Animal Technology Research Center, Agriculture Technology Research Institute, Miaoli 35053, Taiwan; 2Department of Pediatrics, Taipei Hospital, Ministry of Health and Welfare, Taipei 11267, Taiwan; 3School of Medicine, National Yang Ming Chiao Tung University, Taipei 112, Taiwan; 4College of Public Health, China Medical University, Taichung 40402, Taiwan; 5National Institute of Environmental Health Sciences, National Health Research Institutes, Miaoli 35053, Taiwan

**Keywords:** diesel exhaust particles, asthma, ovalbumin, probiotics, allergic disease

## Abstract

Global air pollution and diesel exhaust particles (DEPs) generated by intratracheal instillation aggravate asthma. In this study, we evaluated the effect of probiotics via tracheal- or oral-route administration on allergies or asthma. We continuously perfused rats daily, using the oral and tracheal routes, with approximately 10^6^–10^8^ CFU probiotics, for 4 weeks. During this period, we used OVA-sensitized rats to build the asthma models. We orally or intratracheally administered *Lactobacillus paracasei* 33 (LP33) to the rats, which reduced the number of total inflammatory cells, lymphocytes, and eosinophils in the bronchoalveolar-lavage fluid, the IgE concentration, and the cytokine levels of TH2 cells, but we found no significant difference in the cytokine levels of TH1 cells. LP33 can be used to prevent asthmatic allergic reactions induced by aerosol particles. Nevertheless, the dosage form or use of LP33 needs to be adjusted to reduce the irritation of lung tissues, which may produce lesions of the trachea. We observed that DEP dosage can alleviate emphysema, and that LP33 has a substantial effect on improving or slowing allergic asthma.

## 1. Introduction

Asthma is a chronic pulmonary-inflammatory disease characterized by a large amount of eosinophilic bulb-infiltration and inflammatory reactions in the bronchus, excessive mucus secretion, and obstruction of airflow in the respiratory tract [1]. Various factors aggregate asthma, such as severe air pollution from the diesel exhaust particulates (DEPs) produced by the combustion of diesel fuel [2]. In Taiwan, the main source of suspended particulates is DEPs, which are produced from large vehicles such as buses and trucks. Epidemiological studies have noted an association between exposure to traffic-derived pollutants and lung function in the asthmatic population [3].

DEPs are composed of elemental carbon, polycyclic aromatic hydrocarbons, acidic aerosols, volatile organic compounds, etc. After being inhaled by the respiratory system, DEPs penetrate deep into the alveoli and even into the microvessels of the alveoli, and freely penetrate the cells and tissues of the human body, causing systemic effects, especially a substantial increase in the number of lung and bronchial diseases [4].

Numerous researchers have assessed the effect of particulate pollutants, and particularly of DEPs, on the respiratory system in animal models. In a study of the effect of house-dust extract (HDE) on BALB/c mice, the results showed that DEPs increased HDE-induced airway inflammation, airway mucus-production, oxidative response, and inflammatory-cell infiltration, as well as CXC chemokines at bronchoalveolar-lavage concentrations and airway hyperresponsiveness (AHR) [5]. The variation in exposure to DEPs (low dose: 100 μg/m^3^ DEPs, high dose: 3 mg/m^3^ DEPs) for 1 h per day, 5 days per week, for 3 consecutive months had a varied effect on the numbers of neutrophils and lymphocytes. The high-dose DEP group was more affected than the low-dose DEP and control groups. Additionally, the levels of cytokines IL-5, IL-13, and IFN-γ in the low-dose DEP group were higher than those in the control group, which indicated that long-term DEP exposure may increase AHR, the inflammatory response, and pulmonary fibrosis in mice [5]. Because asthma is a complex disorder, immunization with ovalbumin (OVA) is a classic approach to induce eosinophilic asthma. However, using a single allergen to induce the animal model is not sufficient to reflect all the characteristics of asthma patients. Many studies have reported that asthma associated with neutrophilia was related to bacterial endotoxin, ozone and particulate air-pollution. In the present study, we established eosinophilic asthma through traditional OVA immunization and, on this basis, established neutrophilic asthma by the intratracheal administration of DEPs.

Probiotics are nonpathogenic microorganisms that have been widely used in premedical research for asthma treatments. Various probiotic strains can alter cytokine production in the gut and related lymphoid tissues. Lactic acid bacteria reduce allergic symptoms in mice caused by ovalbumin (OVA), mainly by increasing the T helper 1 (Th1) immune response, inhibiting the T helper 2 (Th2) immune response, and reducing the allergic response. An animal model was used to explore the prevention or improvement effect of allergic asthma with probiotics, showing that feeding probiotics did not affect the food intake or body weight of mice, and improved the bacterial phase in mouse feces and the cecum. *Lactobacillus Paracasei* 33 (LP33) is one of the probiotics that reportedly benefits allergic-disease treatment with immunomodulatory effects in asthmatic animal-models [6].

In animal studies, probiotics have generally been orally administered at doses ranging from 10^6^ to 10^9^ CFU [7,8], or 2 × 10^9^ CFU per mouse [9]. The intranasal administration of probiotics in mice and humans is more effective in regulating airway inflammation [7]. Therefore, with the development of air pollution and the use of probiotics, in this study we aimed to evaluate the effects of DEPs on asthmatic rats with intratracheal or oral probiotic-administration.

## 2. Materials and Methods

### 2.1. Animal and Drug Administration 

#### 2.1.1. Animals

We purchased six-week-old male SD rats from BioLASCO Co., Ltd., Taipei, Taiwan which were allowed to acclimate for a minimum of 1 week before exposure. All the animal experiments and care were approved by the Institutional Animal Care and Use Committee (IACUC) of the Agricultural Technology Research Institute. Rats (weighing 150–250 g) had ad libitum access to rodent chow and water. We set the environment to a 12 h dark/light cycle, with a temperature of 24 ± 2 °C, and 50% ± 20% relative humidity.

#### 2.1.2. Drug Administration

We randomly assigned 48 male SD rats into six groups (*n* = 8 each group) as follows:Group A (IT control group): IT-PBS + IP-challenged PBS + IT PBS,Group B (PO control group): PO-PBS + IP-challenged PBS + IT PBS,Group C (OVA-DEP group): IT-PBS + IP-challenged OVA + IT DEP,Group D (IT Probiotics group): IT-Probiotics (5 × 10^7^ CFU/mL, 0.1 mL/rat) + IP-challenged OVA + IT DEP,Group E (PO Probiotics group): PO-Probiotics (2.0 × 10^7^ CFU/mL, 10 mL/kg/rat) + IP-challenged OVA + IT DEP,Group F: PO-dexamethasone (0.2 mg/kg/rat) + IP-challenged OVA + IT DEP.

Note 1: Intrathecal Injection (IT), per os (P.O).

Note 2: DEP is administrated to rat by intrathecal delivery. 

Note 3: Intrathecal injection is using a microsprayer.

We exposed the rats to OVA for sensitization and challenge; we sensitized and challenged the rats in the control group (Group A and B) using PBS (pH 7.2). On days 1, 14, 21, and 28, we intraperitoneally injected the rats with 1 mL suspension containing 2 mg of freshly prepared OVA and 10% Al (OH)3, in PBS. On day 1, PBS (Groups A, B, and C), probiotics (Groups D and E), and dexamethasone (Group F) were continuously given, either orally or via intratracheal instillation, to the rats for 28 days. On day 29, we administered the rats 1.7% OVA via intratracheal instillation for sensitization, then ultrasonic atomizing intratracheal for 30 min with 1% OVA, for 5 consecutive days. On days 30, 32, and 34, we continuously exposed the rats to PBS (Group A, B) or DEP (Groups C, D, E, and F) intratracheally. After the last exposure to DEPs, within 24 h, we sacrificed the rat, and then we collected the abdominal aortic blood, BALF, and lung tissues (Figure 1). 

### 2.2. Reagents 

We purchased DEPs (SRM 2975, National Institute of Standards and Technology (NIST, Gaithersburg, MD, USA), ovalbumin (OVA), and dexamethasone from Sigma-Aldrich, Saint Louis, MI, USA. The probiotic product (containing at least 2.0 × 10^9^ colony-forming units of *Lactobacillus paracasei* subsp. *paracasei* LP-33) from Taipei, Taiwan, and other ingredients are microcrystalline cellulose, dicalcium phosphate and magnesium stearate. The diesel exhaust particles (DEPs) range in diameter from 0.02 to 0.2 µm. The density is 1.500–1.900 g/cm^3^ at 20 °C. These particles can adsorb over 450 different organic compounds, including mutagenic and carcinogenic polycyclic-aromatic-hydrocarbons. 

### 2.3. Collection and Analysis of Blood Samples

Within 24 h of the final challenge, we sacrificed the rats and collected blood samples. We collected the blood from the abdominal aorta, which we then centrifuged (3500 rpm at 4 °C for 15 min) and stored at −70 °C. We used ELISA kits to determine serum-IgE concentration.

We washed the left lungs three times with 2 mL PBS (6 mL). We centrifuged the BALF at 2000 rpm at 4 °C, for 10 min. We adjusted the cell precipitation to 1 × 10^6^ cells/mL, and we smeared and dried the cells after staining with Liu’s stain. We counted and identified at least 200 cells as eosinophils, neutrophils, or lymphocytes, under a light microscope at 200× magnification. We centrifuged the supernatant of BALF 3000 rpm at 4 °C for 10 min, which we then stored at −70 °C. We used ELISA kits to determine the IL-4 and TNF-α concentrations in the BALF.

### 2.4. Lung Histopathology

We removed the right lung of the rats, which we then fixed with 10% neutral formalin, embedded in paraffin, sectioned into a thickness of 4−6 μm, and stained with hematoxylin and eosin (H&E) for histopathology. We observed the damage and infiltration of inflammatory cells in the lung tissue, using light microscopy.

### 2.5. Statistical Analysis

Experimental results are reported as mean ± standard deviation (SD). We used SPSS software version 22.0 (IBM, Armonk, NY, USA) for data processing. We performed statistical analysis using one-way ANOVA followed by Dunnett’s *t*-test. We considered *p* < 0.05 to be statistically significant.

## 3. Results

### 3.1. Mortality Rate

In this study, we treated rats with probiotics via oral or intratracheal administration every day (106~108 CFU) for four weeks. During this period, the rats were sensitized using ovalbumin (OVA) and the intratracheal administration of diesel exhaust particles (DEPs, Standard Reference Material 2975, with a mean diameter of the area distribution: 11.2 ± 0.1 μm), to accelerate asthma (Table 1 and Table 2). Rats required anesthesia during tracheal administration because, during this period, the animals are susceptible to tracheal or throat occlusion stimulated by the gavage needle and obstruction of the respiratory tract after administration. Therefore, tracheal administration of LP33 poses a risk of death.

### 3.2. Body-Weight Changes

#### Weight Change Caused by OVA/DEP-Allergy Reduced by LP33 Intratracheal Administration

We intratracheally administered probiotics (LP33) to the experimental group for 28 days, and induced by OVA (2 mg/kg, IP) once per week during this period. We monitored body weight before and after the experimental period. During the period, the average weight of each group continued to increase, except for the group of rats treated with dexamethasone via oral administration (Figure 2A,B). According to the literature, long-term administration of dexamethasone inhibits the food intake and body-weight gain of rats.

### 3.3. Levels of Inflammatory Cells in Bronchoalveolar-Lavage Fluid 

#### 3.3.1. Change in Inflammatory Cell Expression in BALF Caused by OVA/DEP-Allergy Reduced by LP33 Intratracheal-Administration 

In terms of cell number, the total number of cells in Group A was 54.56 ± 31.56 × 10^4^, including eosinophils (0.49 ± 0.41 × 10^4^) and lymphocytes (11.69 ± 7.24 × 10^4^). In Group C, we found notable cell-infiltration (162.65 ± 78.62 × 10^4^), including eosinophils (5.74 ± 1.56 × 10^4^) and lymphocytes (14.47 ± 13.86 × 10^4^). The total cell number (*p* = 0.002 < 0.01) and eosinophil number (*p* = 0.000 < 0.001) statistically differed between Groups A and C, suggesting that the OVA/DEP-induced allergic mode increased the expression of inflammatory cells in the BALF. In addition, in Group D (intratracheal administration of LP33) and Group F (oral administration of dexamethasone), the total cell number was 113.67 ± 39.16 × 10^4^, including eosinophils (2.73 ± 1.23 × 10^4^) and lymphocytes (25.06 ± 13.53 × 10^4^). In Group F, the total cell number was 82.93 ± 35.53 × 10^4^, including eosinophils (3.26 ± 2.01 × 10^4^) and lymphocytes (10.55 ± 3.33 × 10^4^). Compared with Group C, Group F showed significantly reduced cell-infiltration (*p* = 0.021 < 0.05). The total cell and eosinophil numbers in Group D were reduced by 30.1% and 52.4%, respectively; in Group F, they were reduced by 49.0% and 43.2%, respectively (Figure 3a–c).

#### 3.3.2. Change in Inflammatory-Cell Expression in BALF Caused by OVA/DEP-Allergy Reduced by LP33 Oral-Administration 

In terms of cell number, the total number of cells in Group B was 33.19 ± 24.28 × 10^4^, including eosinophils (0.31± 0.21 × 10^4^) and lymphocytes (3.20 ± 2.75 × 10^4^). In Group C, significant cell-infiltration was found (162.65 ± 78.62 × 10^4^), including eosinophils (5.74 ±1.56 × 10^4^) and lymphocytes (14.47 ± 13.86 × 10^4^). Total cell numbers (*p* = 0.000 < 0.001), lymphocytes (*p* = 0.027 < 0.05), and eosinophil (*p* < 0.01) were statistically different between Group B and Group C, suggesting that the OVA/DEP-induced allergic mode could increase the expression of inflammatory cells in BALF. In addition, in Group E (oral administration of LP33 group) and Group F (oral administration of dexamethasone group), the cell numbers of Group E were the following: total cell number (72.21 ± 24.06 × 10^4^), eosinophil (3.84 ± 2.88 × 10^4^) and lymphocytes (8.23 ± 4.86 × 10^4^), and the cell numbers of Group F were the following: total cell number (82.93 ± 35.53 × 10^4^), eosinophil (3.26 ± 2.01 × 10^4^) and lymphocytes (10.55± 3.33 × 10^4^). Compared with Group C, both Group E and Group F reduced the total cell numbers and showed significant differences (*p* = 0.003 < 0.05 and *p* = 0.009 < 0.05, respectively). Among them, the total cell number and eosinophil numbers in Group E were reduced by 55.6% and 33.1% respectively, and in Group F they were reduced by 49.0% and 43.2%, respectively (Figure 4a–c).

### 3.4. Expression of Cytokines in BALF

#### 3.4.1. Change in Cytokine Expression in BALF Caused by OVA/DEP-Allergy Reduced by LP33 Intratracheal-Administration 

In Group A, the contents of total protein, IL-4, and TNF-α were 2.5 ± 1.0 mg/mL, 30.7 ± 2.1 pg/mL, and 9.8 ± 6.4 pg/mL, respectively. In Group C, a large amount of cytokines was secreted, including total protein (5.0 ± 1.5 mg/mL), IL-4 (39.3 ± 7.0 pg/mL), and TNF-α (26.1 ± 9.5 pg/mL). Compared with Group C rats, the expression levels of total protein, IL-4, and TNF-α in the BALF of Group A rats were lower, but only the TNF-α expression level in BALF was statistically different (*p* = 0.038 < 0.05). The cytokine expression in Group D rats (intratracheal administration of LP33) was as follows: total protein, 4.5 ± 0.9 mg/mL; IL-4, 23.7 ± 9.1 pg/mL, and TNF-α, 15.9 ± 9.3 pg/mL. The cytokine expression of the Group F rats (oral administration of dexamethasone) was 4.9 ± 2.7 mg/mL total protein, 32.2 ±7.1 pg/mL IL-4, and 16.4 ± 2.4 pg/mL TNF-α. Compared with Group C, only IL-4 in Group D was significantly reduced (*p* = 0.004 < 0.01). The IL-4- and TNF-α-levels in Group D were reduced by 39.7% and 39.1%, respectively; in Group F, they were reduced by 18.1% and 37.2%, respectively (Figure 5a–c).

#### 3.4.2. Change in Cytokine Expression in BALF Caused by OVA/DEP-Allergy Reduced by LP33 Oral-Administration 

We used ELISA to evaluate the levels of total protein, IL-4, and TNF-α. In Group B, the total protein was 3.2 ± 1.0 mg/mL, IL-4 was 32.8 ± 5.0 pg/mL, and TNF-α was 20.1 ± 10.9 pg/mL. In Group C, a large amount of cytokines was secreted, including total protein (5.0 ± 1.5 mg/mL), IL-4 (39.3 ± 7.0 pg/mL), and TNF-α (26.1 ± 9.5 pg/mL). Compared with Group C, the expression levels of total protein, IL-4, and TNF-α in the BALF of Group B were lower than in Group C, but the differences were not statistically significant (*p* > 0.05). The cytokine expression of Group E was as follows: total protein = 3.8 ± 0.9 mg/mL, IL-4 = 28.8 ± 5.5 pg/mL, and TNF-α = 17.8 ± 4.0 pg/mL. The cytokine expression in the Group F rats was total protein = 4.9 ± 2.7 mg/mL, IL-4 = 32.2 ± 7.1 pg/mL, and TNF-α = 16.4 ± 2.4 pg/mL. Compared with Group C, the IL-4 expression in Group E was lower (*p* = 0.015 < 0.05), and the TNF-α expression was reduced in Group F (*p* = 0.037 < 0.05). The IL-4 and TNF-α levels in Group E were reduced by 26.7% and 31.8%. respectively; in Group F, they were reduced by 18.1% and 37.2%, respectively (Figure 6a–c).

### 3.5. Expression of Total IgE-Concentration in Rat Serum

#### 3.5.1. IgE Concentration in Rat Serum Caused by OVA/DEP-Allergy Reduced by LP33 Intratracheal-Administration 

We used ELISA to evaluate the expression level of IgE. The rat-serum-IgE levels in Groups A and C were 25.0 ± 7.4 and 84.7 ± 49.5 ng/mL, respectively. The concentration of IgE in Group A was lower than that in Group C, but this difference was not statistically significant (*p* > 0.05). In Groups D and F, the serum-IgE concentrations were 41.0 ± 19.2 and 30.5± 19.2 ng/mL, respectively; compared with Group C, the IgE concentration was lower in these groups by 51.6% and 64.0%, respectively. The Group F serum-IgE concentration was significantly different from that of Group C (*p* = 0.035 < 0.05) (Figure 7A).

#### 3.5.2. IgE Concentration in Rat Serum Caused by OVA/DEP-Allergy Reduced by LP33 Oral-Administration 

We used ELISA to evaluate the expression level of IgE. The rat-serum-IgE level in Group B and C rats was 32.2 ± 12.0 and 84.7 ± 49.5 ng/mL, respectively. The concentration of IgE in Group B was statistically significantly lower than in Group C (*p* = 0.043 < 0.05). In Groups E and F, the serum-IgE concentrations were 55.0 ± 32.8 and 30.5± 19.2 ng/mL, respectively, which were lower than in Group C by 35.1% and 64.0%, respectively. The difference between Groups C and F was significant (*p* = 0.028 < 0.05) (Figure 7B).

### 3.6. Histopathology Examination in Rat Lungs

#### 3.6.1. Allergy Reduced by LP33 Intratracheal-Administration 

In Group A (Figure 8A), we observed multifocal emphysema (lesion mean-score: 1.29 ± 0.76), diffuse mononuclear-cell infiltration (lesion mean-score: 1.71 ± 0.76), and multifocal chronic-granulomatous inflammation (lesion mean-score: 0.86 ± 0.90) in the alveoli space. In Group C (Figure 8B), we observed multifocal emphysema (lesion mean-score: 2.83 ± 0.41), diffuse mononuclear-cell infiltration (lesion mean-score: 3.33 ± 0.52), and multifocal chronic-granulomatous inflammation (lesion mean-score: 3.00 ± 1.10) in the alveoli space. In Group D (Figure 8C), we identified multifocal emphysema (lesion mean-score: 2.67 ± 0.82), diffuse mononuclear-cell infiltration (lesion mean-score: 3.57 ± 0.53), and multifocal chronic-granulomatous inflammation (lesion mean-score: 2.43 ± 0.53) in the alveoli space. In Group F (Figure 8D), we observed multifocal emphysema (lesion mean-score: 2.29 ± 0.49), diffuse mononuclear-cell infiltration (lesion mean-score: 3.00 ± 0.00), and multifocal chronic-granulomatous inflammation (lesion mean-score: 1.43 ± 0.79) in the alveoli space.

#### 3.6.2. Allergy Reduced by LP33 Oral Administration

In Group B (Figure 9A), we observed multifocal emphysema (lesion mean-score: 0.88 ± 0.99), diffuse mononuclear-cell infiltration (lesion mean-score: 1.75 ± 0.71), and multifocal chronic-granulomatous inflammation (lesion mean-score: 0.75 ± 1.04) in the alveolar space. In Group C (Figure 9B), we noted multifocal emphysema (lesion mean-score: 2.83 ± 0.41), diffuse mononuclear-cell infiltration (lesion mean-score: 3.33 ± 0.52), and multifocal chronic-granulomatous inflammation (lesion mean-score: 3.00 ± 1.10) in the alveoli space. In Group E (Figure 9C), we found multifocal emphysema (lesion mean-score: 2.71 ± 0.49), diffuse mononuclear-cell infiltration (lesion mean-score: 3.67 ± 0.52), and multifocal chronic-granulomatous inflammation (lesion mean-score: 4.17 ± 0.41) in the alveoli space. In Group F (Figure 9D), we found multifocal emphysema (lesion mean-score: 2.29 ± 0.49), diffuse mononuclear-cell infiltration (lesion mean-score: 3.00 ± 0.00), and multifocal chronic-granulomatous inflammation (lesion mean-score: 1.43 ± 0.79) in the alveolar space.

## 4. Discussion

Asthma is an allergic disease that causes lung inflammation, with common symptoms including intermittent dyspnea, chest tightness, and cough [10]. The pathological characteristics of asthma include airway inflammation, reversible airflow-obstruction, and obvious changes in respiratory-tract structure, such as thickened bronchial-epithelium, mucus hypersecretion, bronchial smooth-muscle hyperplasia, and lung inflammation [11]. Many clinical drugs are available for the treatment of asthma, such as inhaled corticosteroids, β2 agonists, theophylline, and leukotriene antagonists. Among them, inhaled corticosteroids are the most effective at controlling asthma and improving pulmonary function [12]. However, whether these drugs are used alone or in combination with other treatments, they do not cure the disease. 

In 2016, the World Health Organization indicated that indoor and outdoor air-pollution can damage the respiratory tract, including through acute respiratory-tract infection, chronic obstructive pulmonary disease, and exacerbated allergic-airway response, and is strongly related to cardiovascular disease and cancer [13]. Recently, allergic diseases were found to be related to the presence of probiotics in the intestines; many researchers have used probiotics to improve allergic diseases caused by immune-system imbalance, such as anaphylactic rhinitis, dermatitis, inflammatory bowel disease, etc. [14,15,16]. These probiotics are mostly lactic acid bacteria, such as Lactobacillus, Lactococcus, Leuconostoc, Pediococcus, etc. [17]. The probiotic we used in this study was *Lactobacillus paracasei* 33(LP33). Many animal studies have confirmed that *Lactobacillus paracasei* can inhibit the synthesis of specific immunoglobulin E (IgE) and stimulate interleukin-12 (IL-12) production [18,19,20], which is thought to protect against, prevent, and improve allergic diseases.

We monitored the body weight of the rats before and after the experiment. During this period, the average weight of each group continued to increase, except for that of the rats in the group orally treated with dexamethasone, which agrees with the result reported in the literature [21]. Next, we explored whether LP33 could reduce or improve the inflammatory immune-response caused by allergies, such as the inflammatory response, the concentration of IgE in the serum, and the secretion of inflammatory mediators and cytokines. The expression of inflammatory cells in the BALF, including total cells, eosinophils, and lymphocytes, were lower in the rats orally or intratracheally administered with LP33 than in the PC-group rats. Furthermore, we analyzed the cytokine-expression levels in the BALF. The TH2 cells secrete IL-4, which inhibits TH1-cell activity, stimulates B cells to produce IgE, and stimulates the growth of mast cells, to induce the proliferation of VCAM-1 and eosinophils. Our results showed that LP33 inhibited IL-4, promoted the transition of precursor T cells to TH1, and reduced the inflammatory response. However, if the TH1 cytokine TNF-α is used as a symbolic inflammation-marker, the oral and intratracheal administration of LP33 failed to reach our expectations. We speculated that the reason for this finding might be related to the different times of TH1/TH2 cytokine production.

In the future, we could monitor TH1/TH2 cytokine levels at different time-points during the experiment or observe the changes in the growth and decline in these levels. In addition, we added air particles to accelerate the induction of asthma in this experiment; this method slightly differs from that used in previous studies. Possibly, we may explore additional TH1/TH2 cytokines and mechanisms. In the histopathological results, we identified mononuclear-inflammatory-cell infiltrate, granulomatous inflammation, and emphysematous lesions. According to the lesion severity score, both OVA and DEP stimulate lesions in the lung tissue, and dexamethasone can effectively reduce these lesions. The typical symptoms of general allergic reactions are bronchial epithelial hyperplasia, pulmonary inflammation, excessive mucus secretion, and bronchial smooth-muscle hyperplasia [22]. Therefore, we speculate that long-term saline or LP33 administration to the trachea will cause granuloma, and DEPs can cause pulmonary emphysema, resulting in obstructive pulmonary disease.

## 5. Conclusions

Based on our results, whether *Lactobacillus paracasei* 33 (LP33) were administered orally or tracheally, we observed its tendency to reduce the total number of inflammatory cells, lymphocytes, and eosinophils in BALF. A reduction in the expression level of TH2 cytokines and IgE was not evident in TH1. We think that LP33 can be used to prevent asthmatic allergic reactions induced by air particles, but the dosage form or use of LP33 needs to be adjusted to reduce pulmonary irritation. In the future, we could try to change the DEP dose to reduce the lesions of pulmonary emphysema so we can better observe the effect of LP33 on improving or alleviating allergic asthma.

## Figures and Tables

**Figure 1 nutrients-14-05219-f001:**
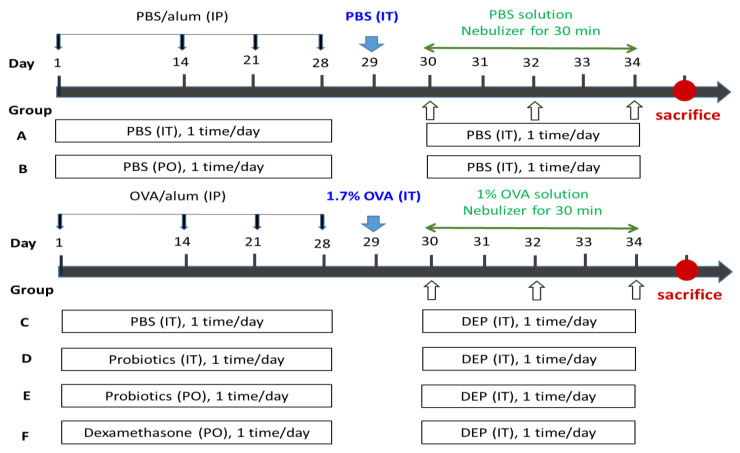
Flow chart of the animal experiment: six groups of rats were orally or intratracheally administered instillation probiotics or PBS or dexamethasone for 28 days continuously. Rats were intraperitoneally injected with OVA and Al (OH)3 in PBS on days 1, 14, 21, and 28. On day 29, rats were given 1.7% OVA via intratracheal instillation for sensitization, then ultrasonic-atomizing for 30 min with 1% OVA for 5 consecutive days. On days 30, 32, and 34, rats were continuously exposed to DEPs or PBS by intratracheal administration. Within 24 of the last exposure to DEP, the rats were sacrificed. PBS, phosphate-buffered saline; alum, aluminum hydroxide; OVA, ovalbumin; DEPs, diesel exhaust particles; IP, intraperitoneal; PO, per oral; IT, intratracheal instillation.

**Figure 2 nutrients-14-05219-f002:**
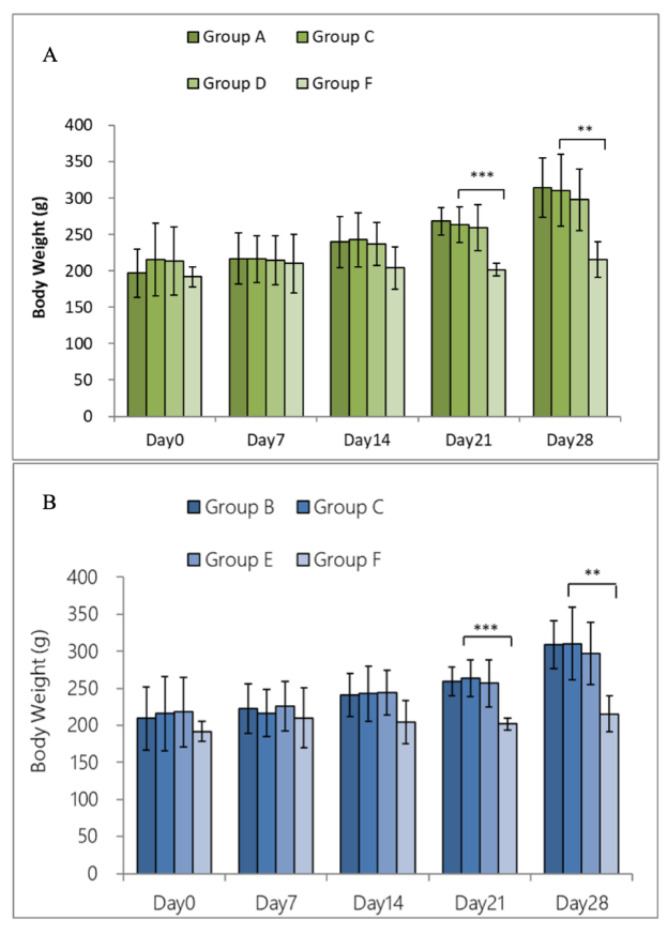
Effect of (**A**) intratracheal and (**B**) oral-probiotic-*Lactobacillus paracasei* LP33 administration on body-weight change induced by OVA/DEP in asthmatic rats. Mean ± SD values are shown for the four groups of rats. Each group contained 8 rats. Mean statistical significance compared with asthma group using Dunnett’s *t*-test; ** *p* < 0.01, *** *p* < 0.001.

**Figure 3 nutrients-14-05219-f003:**
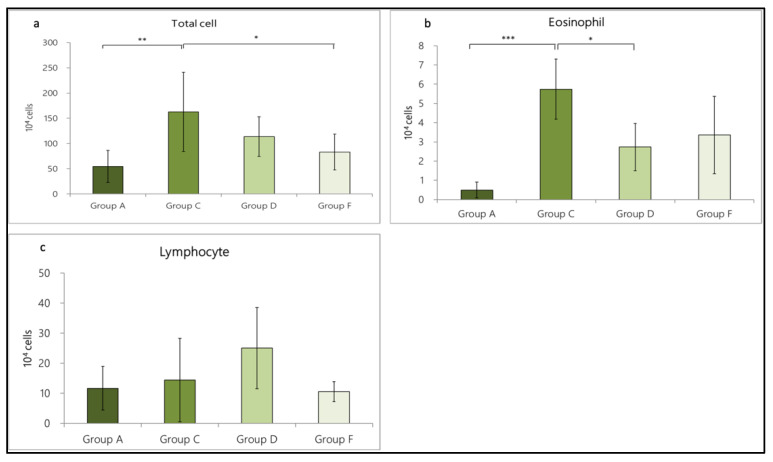
Effect of intratracheal-probiotic-*Lactobacillus paracasei* LP33 administration on inflammatory-cell proliferation induced by OVA/DEP in asthmatic rats: (**a**) total cell, (**b**) eosinophil, and (**c**) lymphocyte counts. Mean ± SD values are shown for the four groups of rats. Each group contained 8 rats. Statistical significance compared with asthma group using Dunnett *t*-test: * *p* < 0.05, ** *p* < 0.01, and *** *p* < 0.001.

**Figure 4 nutrients-14-05219-f004:**
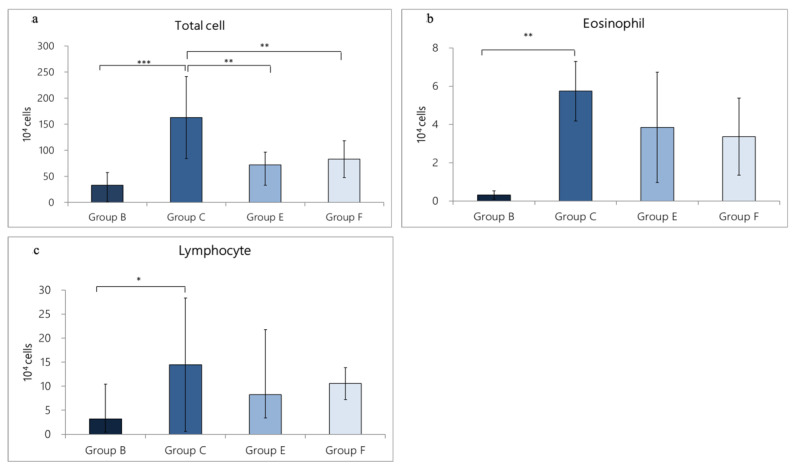
Effect of oral-probiotic-*Lactobacillus paracasei* LP33 administration on inflammatory-cell proliferation induced by OVA/DEP, in asthmatic rats: (**a**) total cell, (**b**) eosinophil, and (**c**) lymphocyte counts. Mean ± SD values for four groups of rats. Each group contained 8 rats. Statistical significance compared with asthma group using Dunnett’s *t*-test: * *p* < 0.05, ** *p* < 0.01, and *** *p* < 0.001.

**Figure 5 nutrients-14-05219-f005:**
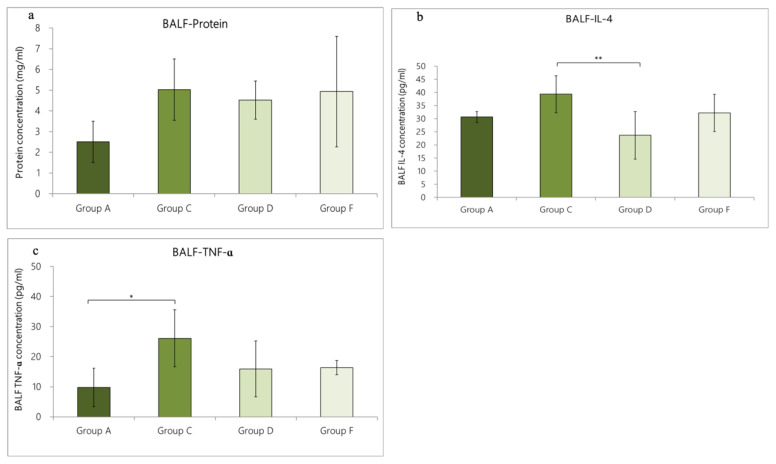
Effect of intratracheal-probiotic-*Lactobacillus paracasei* LP33 administration on cytokine levels in BALF induced by OVA/DEP, in asthmatic rats: (**a**) protein, (**b**) IL-4, and (**c**) TNF- α. Mean ± SD values for four groups of rats. Each group contained 8 rats. Statistical significance compared with asthma group using Dunnett’s *t*-test: * *p* < 0.05 and ** *p* < 0.01.

**Figure 6 nutrients-14-05219-f006:**
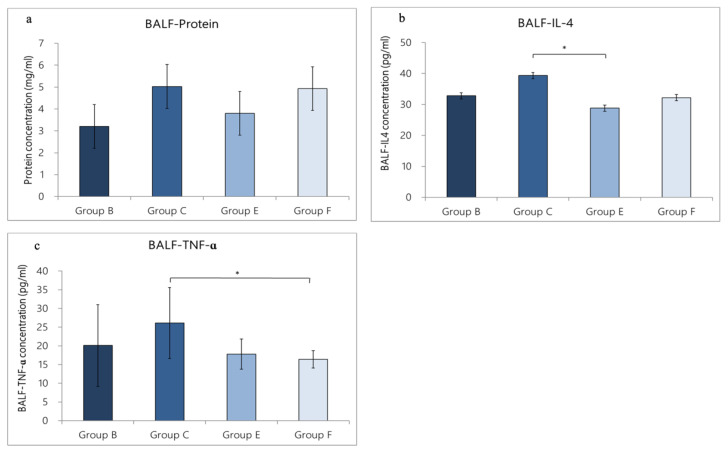
Effect of oral-probiotic-*Lactobacillus paracasei* LP33 administration on BALF cytokine-levels induced by OVA/DEP, in asthmatic rats: (**a**) protein, (**b**) IL-4, and (**c**) TNF-α. Mean ± SD values of four groups of rats. Each group contained 8 rats. Statistical significance compared with asthma group using Dunnett’s *t*-test: * *p* < 0.05.

**Figure 7 nutrients-14-05219-f007:**
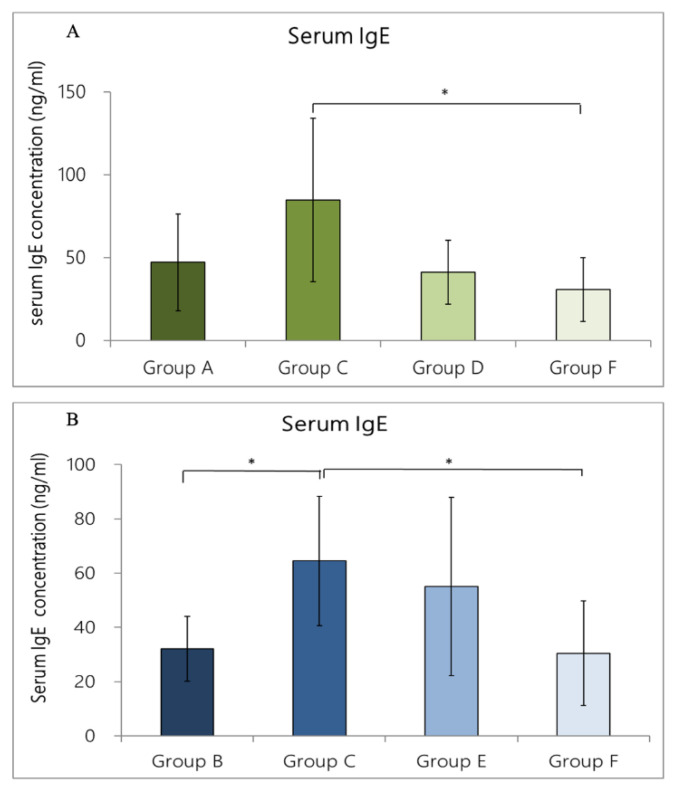
Effect of (**A**) intratracheal and (**B**) oral-probiotic-*Lactobacillus paracasei* LP33 administration on serum-IgE concentrations induced by OVA/DEP, in asthmatic rats. Mean ± SD values for four groups of rats. Each group contained 8 rats. * Statistical significance compared with asthma group using Dunnett’s *t*-test: * *p* < 0.05.

**Figure 8 nutrients-14-05219-f008:**
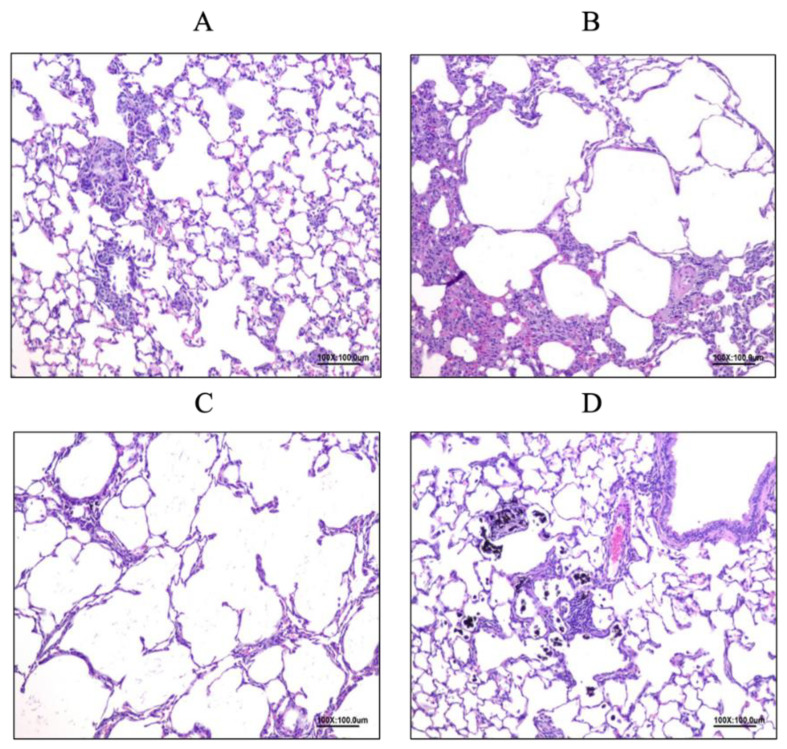
Effect of intratracheal-probiotic-*Lactobacillus paracasei* LP33 administration on histopathological changes in lung tissues induced by Group C (OVA-DEP group) in asthmatic rats: (**A**) granuloma, multiple localized, mild, lung; (**B**) emphysema, alveolar cavity, multiple local, moderate, lung; (**C**) emphysema, alveolar cavity, multiple local, moderate, lung; (**D**) granuloma, multiple localized, mild, lung. Magnification for all: 100×.

**Figure 9 nutrients-14-05219-f009:**
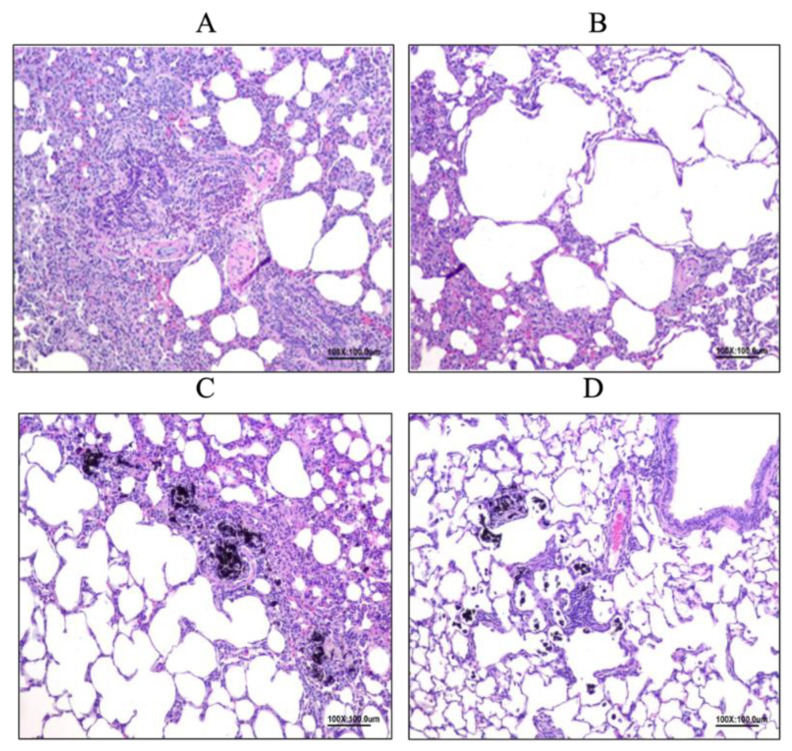
Effect of oral-probiotic-*Lactobacillus paracasei* LP33 administration on histopathological changes in lung tissues induced by Group E (PO Probiotics group) in asthmatic rats: (**A**) infiltration, monocytes, spread, mild, lung; (**B**) emphysema, alveolar cavity, multiple local, moderate, lung; (**C**) granuloma, multiple local, moderate, lung; (**D**) granuloma, multiple localized, mild, lung. Magnification: 100×.

**Table 1 nutrients-14-05219-t001:** Effect of intratracheal-probiotic-*Lactobacillus paracasei* LP33 administration on mortality induced by OVA/DEP, in asthmatic rats.

Group ^1^	Total Incidence (N/N ^2^)	Cause
Group A	1/8	One rat died after intratracheal PBS administration
Group C	2/8	Two rats died after OVA induction
Group D	2/8	One rat died after OVA induction; one rat died after intratracheal-probiotic administration
Group F	0/8	

^1^ Group A: IT-PBS + IP-challenged PBS+ Intratracheal PBS, Group C: IT-PBS + IP-challenged OVA+ Intratracheal DEP, Group D: IT-probiotics (LP33, 5 × 10^7^ CFU/mL, 0.1 mL/rat) + IP-challenged OVA + Intratracheal DEP, Group F: PO-dexamethasone 0.2 mg/kg/rat) + IP-challenged OVA + intratracheal DEP. ^2^ N/N: Total number of rats found dead or moribund animals/total number of animals.

**Table 2 nutrients-14-05219-t002:** Effect of oral-probiotic-*Lactobacillus paracasei* LP33 administration on mortality induced by OVA/DEP, in asthmatic rats.

Group ^1^	Total Incidence (N/N ^2^)	Cause
Group B	0/8	
Group C	2/8	Two rats died after OVA induction
Group E	1/8	One rat died after OVA induction
Group F	0/8	

^1^ Group B: PO-PBS + IP-challenged PBS + intratracheal PBS, Group C: POPBS + IP-challenged OVA + intratracheal DEP, Group E: IT-Probiotics (LP33, 2.0 × 10^7^ CFU/mL, 10 mL/kg) + IP-challenged OVA + intratracheal DEP, and Group F: PO-dexamethasone (0.2 mg/kg/rat) + IP-challenged OVA + intratracheal DEP. ^2^ N/N: Total number of rats found dead or moribund animals/total number of animals.

## Data Availability

Not applicable.

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
