# Peer review of "Probiotics’ Efficacy in Preventing Asthmatic Allergic Reaction Induced by Air Particles: An Animal Study"

_nutrients, 2022, doi:10.3390/nu14245219_

Round 1

Reviewer 1 Report

The authors investigated the effect of probiotics via tracheal or oral route administration in DEP-exposed allergic asthmatic rats. They showed that oral or intratracheal treatment of  LP33 reduced the infiltration of inflammatory cells into the lungs, IgE secretion, and Th2 cytokines in BALF. This study is interested, but I seem this article has a  problem with the experimental design.

1.    The authors compared four groups (B, C, D, and F) to evaluate the effects of oral administration of LP33. You compared the PBS(PO)+DEP/OVA to PBS(IT)+DEP/OVA, but you should confirm the differences to the PBS(PO)+DEP/OVA. Also, without a comparison to PBS (PO or IT)+OVA, it is unclear whether LP33 treatment is effective in exacerbating allergic inflammation due to DEP exposure. Thus, it is not clear in this experiment whether there was an anti-inflammatory effect of oral treatment of LP33. Also, the difference between the ameliorative effect of OVA/DEP and OVA alone remains unclear.

Introduction

1.    The authors need to explain why they chose to use intratracheal administration as well as oral administration as a method of probiotics treatment.

2.    The authors do not explain in the Introduction why they chose LP33. You had better move the sentences in the Discussion (P12. L347-P13. L355).

Major comments
Methods

1.    The authors should specify the age of the rats.

2.    Please add an explanation of the abbreviations PO and IT in the text. The authors do not provide any information on the probiotics used (origin, volume). The authors should also describe in detail the method of oral or intratracheal administration and the dosage of dexamethasone.

3.    The authors should describe the exposure dose and administration method of DEP. It is doubtful that the effects of OVA, DEP, and probiotics are accurately detected in an experimental design in which deaths occur with PBS alone. It is also undeniable that individuals died after OVA inhalation may also be due to damage of intratracheal administration.

Results

4.    Except for the Abstract, the authors the word mentioned "probiotics (LP33)" for the first time in 3.2 in Results, but you should describe in the "Reagent".

5.    The authors used 10 animals/group in Figures 2-5, but 8 animals/group were used in Table 1. If it is a different experiment, the authors should explain about it.

6.    The authors mentioned that DEP administration occurs multifocal emphysema in OVA sensitized mice, but in general, emphysematous changes are rarely observed in allergic asthma models.

7.    The authors mentioned that Figure 6.2a is the control group (PBS(PO)+PBS inhalation), but lung emphysema is occurred. The legend in Figure 6 should describe group names.

Minor comments:

P6. L201. eosinocyteeosinophil

P13. L378. OVEOVA

Author Response

Dear Editors and Reviewers,

Thank you for your useful comments. We made changes in this manuscript accordingly and hope that the current revised version fulfills your requirements. Detailed corrections are listed below, and the manuscript revisions are highlighted in red.

We sincerely hope that the revised version of our manuscript will be considered for eventual publication in your prestigious journal.

Reply to Reviewer 1

The authors investigated the effect of probiotics via tracheal or oral route administration in DEP-exposed allergic asthmatic rats. They showed that oral or intratracheal treatment of  LP33 reduced the infiltration of inflammatory cells into the lungs, IgE secretion, and Th2 cytokines in BALF. This study is interested, but I seem this article has a problem with the experimental design.

 Q1.    The authors compared four groups (B, C, D, and F) to evaluate the effects of oral administration of LP33. You compared the PBS(PO)+DEP/OVA to PBS(IT)+DEP/OVA, but you should confirm the differences to the PBS(PO)+DEP/OVA. Also, without a comparison to PBS (PO or IT)+OVA, it is unclear whether LP33 treatment is effective in exacerbating allergic inflammation due to DEP exposure. Thus, it is not clear in this experiment whether there was an anti-inflammatory effect of oral treatment of LP33. Also, the difference between the ameliorative effect of OVA/DEP and OVA alone remains unclear.

 ANS:

Thanks for your insightful opinions. In this study, we focused to investigate the preventing asthmatic allergic reaction by different routes administration of probiotics. Animal models for simulating human asthma and exposure to ambient air pollution. The dexamethasone is used as positive control. The results were showed IT probiotic group D in eosinophil, IL-4 in BALF and multifocal emphysema, multifocal chronic granulomatous inflammation of histopathology examination are reduced compared to group C. The results were showed PO probiotic group E in total cell, IL-4 in BALFand multifocal emphysema of histopathology examination are reduced compared to group C.

Introduction

  1. The authors need to explain why they chose to use intratracheal administration as well as oral administration as a method of probiotics treatment.

ANS:

Thanks for your insightful opinions. Most asthma attacks are triggered by inhaled DEP. Using inhaled corticosteroid for asthma has less side effect than oral steroid. Direct contract with airway mucosa may have better treatment response and may change local microbiota. So, we want to know if intra-tracheal administration of probiotics had better effect than oral administration. We may develop new drug for asthma.      

  1. The authors do not explain in the Introduction why they chose LP33. You had better move the sentences in the Discussion (P12. L347-P13. L355).
    ANS:

Thanks for your insightful opinions.

We already add the additional information related to LP33 in the introduction. (P02.L60-L61). We use LP33 because of its benefits in allergic disease treatment with immunomodulatory effects in asthmatic animal models based on previous literature. In addition, we also deleted the requested sentences in the discussion.

Major comments
Methods

  1. The authors should specify the age of the rats.

ANS: Thanks for your insightful opinions. Animals entered at six week-old. It has been supplemented in the 2.1.1 sentence

  1. Please add an explanation of the abbreviations PO and IT in the text. The authors do not provide any information on the probiotics used (origin, volume). The authors should also describe in detail the method of oral or intratracheal administration and the dosage of dexamethasone.

ANS:

Thanks for your insightful opinions. Probiotic product (containing at least 2.0×109 colony forming units of Lactobacillus paracasei subsp. paracasei LP-33), and other ingredient are microcrystalline cellulose, dicalcium phosphate and magnesium stearate. Dexamethasone of dosing has described in 2.1.2. Group F: PO-dexamethasone (0.2 mg/kg/rat).

  1. The authors should describe the exposure dose and administration method of DEP. It is doubtful that the effects of OVA, DEP, and probiotics are accurately detected in an experimental design in which deaths occur with PBS alone. It is also undeniable that individuals died after OVA inhalation may also be due to damage of intratracheal administration.

ANS:

Administration method of DEP by i.t administration ( 1 mL/kg). We will find the preventing asthmatic allergic reaction effect of probiotics in the experiment, the OVA and DEP is used to induce allergic reaction in animal. There are some animals were sensitized by intratracheal administration of OVA and caused death.

Results

  1. Except for the Abstract, the authors the word mentioned "probiotics (LP33)" for the first time in 3.2 in Results, but you should describe in the "Reagent".

ANS: Added explanation in 2.2 Reagent sentence.

2.2. Reagents

We purchased DEPs (SRM 2975, National Institute of Standards Technology (NIST)), ovalbumin (OVA), and dexamethasone from Sigma-Aldrich. Probiotic product (containing at least 2.0×109 colony forming units of Lactobacillus paracasei subsp. paracasei LP-33), and other ingredient are microcrystalline cellulose, dicalcium phosphate and magnesium stearate.

  1. The authors used 10 animals/group in Figures 2-5, but 8 animals/group were used in Table 1. If it is a different experiment, the authors should explain about it.

ANS: There were 8 animals/group. Each group contained 10 rats were misplanted. Corrected to “Each group contained 8 rats”. Thanks for your insightful opinions.

  1. The authors mentioned that DEP administration occurs multifocal emphysema in OVA sensitized mice, but in general, emphysematous changes are rarely observed in allergic asthma models.

ANS: The paper (Dried Yeast Extracts Curtails Pulmonary Oxidative Stress, Inflammation and Tissue Destruction in a Model of Experimental Emphysema) was published in 2019 that mice were induced pulmonary emphysema by cigarette smoke (CS) and ovalbumin (OVA). The results indicated the yeast extracts may therapeutically ameliorate oxidative stress and inflammatory tissue destruction in emphysematous diseases. The paper is provided for your reference.

  1. The authors mentioned that Figure 6.2a is the control group (PBS(PO)+PBS inhalation), but lung emphysema is occurred. The legend in Figure 6 should describe group names.

 ANS: It has been corrected. Thanks for your insightful opinions.

Minor comments:

P6. L201. eosinocyte→eosinophil

P13. L378. OVE→OVA

ANS:

Thanks for your insightful opinions. It has been corrected.

Reviewer 2 Report

The authors present an animal study concerning Probiotics’ efficacy in preventing asthmatic allergic reactions induced by air particles. The authors conclude that Lactobacillus paracasei 33 (LP33) reduces the total number of inflammatory cells, lymphocytes, and eosinophils in BALF. Considering that reducing the expression level of TH2 cytokines and IgE was not evident in TH1, the authors think that LP33 can be used to prevent asthmatic allergic reactions induced by air particles, but the dosage form or use of LP33 needs to be adjusted to reduce pulmonary irritation. This last point should be carefully pounded if different doses and forms of DEP and probiotic LP33 could have adverse effects and lesions on the lung tissue, on the way to improve allergic asthma outcomes. 

Author Response

The authors present an animal study concerning Probiotics’ efficacy in preventing asthmatic allergic reactions induced by air particles. The authors conclude that Lactobacillus paracasei 33 (LP33) reduces the total number of inflammatory cells, lymphocytes, and eosinophils in BALF. Considering that reducing the expression level of TH2 cytokines and IgE was not evident in TH1, the authors think that LP33 can be used to prevent asthmatic allergic reactions induced by air particles, but the dosage form or use of LP33 needs to be adjusted to reduce pulmonary irritation. This last point should be carefully pounded if different doses and forms of DEP and probiotic LP33 could have adverse effects and lesions on the lung tissue, on the way to improve allergic asthma outcomes. 

ANS:
Thanks for your suggestion. In this study, we focused to investigate the preventing asthmatic allergic reaction by different routes administration of probiotics. In the future, we will further design the safety study in different dose to explore the efficacy and adverse effect.

Round 2

Reviewer 1 Report

The authors have generally responded to the comments.

The authors had better to describe overall ‘a DEP-administerd (or exposed) allergic asthmatic model’, but not ‘DEP-induced allergic asthma’ . Because you did not indicate a difference between OVA and DEP/OVA and answered that animal models in this study is 'for simulating human asthma and exposure to ambient air pollution'.

Author Response

Dear Editors and Reviewers,

Thank you for your useful comments. We made changes in this manuscript accordingly and hope that the current revised version fulfills your requirements. Detailed corrections are listed below and the manuscript revisions are highlighted in red.

We sincerely hope that the revised version of our manuscript will be considered for eventual publication in your prestigious journal.

Reply to Reviewer 1

The authors have generally responded to the comments.

The authors had better to describe overall ‘a DEP-administerd (or exposed) allergic asthmatic model’, but not ‘DEP-induced allergic asthma’ . Because you did not indicate a difference between OVA and DEP/OVA and answered that animal models in this study is 'for simulating human asthma and exposure to ambient air pollution'.

ANS:

Thanks for your insightful opinions.

Because the asthma is a complex disorder. Immunization with ovalbumin (OVA) is a classic approach to induce eosinophilic asthma. However, using single allergen to induce animal model is not sufficient to reflect all the characteristics of asthma patients. Many studies have reported that asthma associated with neutrophilia was related to bacterial endotoxin, ozone and particulate air pollution. In the present study, we established eosinophilic asthma through traditional OVA immunization and, on this basis, established neutrophilic asthma by intratracheal administration of DEPs. In brief, we use OVA and DEP to induced mixed asthma animal model.

We also modify the manuscript. Please see page 2 in the manuscript.

 2.2. Reagents
We purchased DEPs (SRM 2975, National Institute of Standards Technology
(NIST)), ovalbumin (OVA), and dexamethasone from Sigma-Aldrich. Probiotic product (containing at least 2.0×10 9 colony forming units of Lactobacillus paracasei subsp. paracasei LP-33), and other ingredient are microcrystalline cellulose, dicalcium phosphate and magnesium stearate. Diesel exhaust particle (DEP) range in diameter from 0.02 to 0.2 µm. The density 1.500 - 1.900 g/cm 3 at 20 ℃. These particles can adsorb over 450 different organic compounds, including mutagenic and carcinogenic
polycyclic aromatic hydrocarbons.